# Chinese herbal therapy in the management of rhinosinusitis—A systematic review and meta-analysis

Jing Cui[1], Wenmin Lin[2,3], Brian H. May[1], Qiulan Luo[2,3], Christopher Worsnop[4], Anthony Lin Zhang[1], Xinfeng Guo[2], Chuanjian Lu[2], Yunying Li[2,3], Charlie C. Xue[1,2]*

1 China-Australia International Research Centre for Chinese Medicine, School of Health and Biomedical Sciences, RMIT University, Bundoora, Victoria, Australia, 2 Guangdong Provincial Academy of Chinese Medical Sciences, Guangdong Provincial Hospital of Chinese Medicine, and The Second Clinical College, Guangzhou University of Chinese Medicine, Guangzhou, China, 3 Department of Otolaryngology, Guangdong Provincial Hospital of Chinese Medicine, Guangzhou, China, 4 Department of Respiratory Medicine, Austin Health, Heidelberg, Victoria, Australia

* charlie.xue@rmit.edu.au

**Data Availability Statement:** All relevant data are within the manuscript and its Supporting Information files.

## Abstract

This systematic review aims to assess the effects and safety of Chinese herbal medicines (CHMs) in the management of rhinosinusitis (RS); inform clinicians of the current state of the evidence; identify the best available evidence; and suggest further directions for research. Five English and four Chinese language databases, and four clinical trial registries were searched. Eligible studies were randomised controlled trials (RCTs). Participants were diagnosed with RS based on established criteria. Test interventions were CHMs administered orally and/or nasally, excluding injections and displacement techniques. Control interventions included placebos, no additional treatment, and conventional non-invasive treatments including pharmacotherapies and/or nasal irrigation, and/or inhalations. Polyposis and post-surgical recovery were excluded. Outcomes were Sino-Nasal Outcome Test (SNOT), visual analogue scales (VAS), Lund-Mackay computed tomography score (LM), Lund-Kennedy Endoscopic score (LK), Mucociliary transport time (MTT), Mucociliary transport rate (MTR), quality of life and adverse events (AEs). Risk of bias used the Cochrane tool. Meta-analysis in Review Manager 5.4.1 used random effects for mean difference (MD) or risk ratio (RR) with 95% confidence intervals. Heterogeneity was assessed as $I^2$. Thirty-four RCTs were included, 30 of chronic RS (CRS) and four of acute RS (ARS). These enrolled 3,752 participants. Five RCTs blinded participants. For CRS, comparisons with placebo showed greater improvements in the CHM groups for SNOT-20 and VAS-TNS (total nasal symptoms). Blinded comparisons with pharmacotherapies showed no differences between groups in the degree of improvement for SNOT-20, VAS-TNS, and LM, suggesting these CHMs had similar effects, at least in the short term. In ARS, pooled results found improved scores on VAS-TNS and LK suggesting a benefit for combining these CHMs with pharmacotherapies. Limitations included inadequacies in study design and methodological reporting, and insufficient reporting of AEs. Heterogeneity in some pooled results precluded strong conclusions. Further well-designed studies are needed to test whether the results are replicable.

**Funding:** Funding support was provided by the China-Australia International Research Centre for Chinese Medicine (CAIRCCM) - a joint initiative of RMIT University, Australia and Guangdong Provincial Academy of Chinese Medical Sciences, China, and the Foundation for Chinese Medicine and Technology Research of Guangdong Provincial Hospital of Chinese Medicine (2017KT1820, 2016KT1571). The funders had no role in study design, data collection and analysis, decision to publish, or preparation of the manuscript.

**Competing interests:** The authors have declared that no competing interests exist.

**Systematic review registration number:** PROSPERO (CRD42019119586).

## Introduction

Rhinosinusitis (RS) is an inflammation of the paranasal sinuses and nasal cavity [1, 2]. When less than four weeks in duration, it is classified as acute RS (ARS) and when more than 12 weeks in duration it is chronic RS (CRS) [2]. Surveys have estimated RS affected 10.9% of the European adult population [3] and 12.1% of the American population [4]. In eastern Asia, CRS prevalence in South Korea was 6.95% based on a survey plus physical examination [5] and 10.78% in a symptom-based survey [6]; and was 8.0% (4.8–9.7%) in a survey of seven Chinese cities [7]. Rhinosinusitis was the fifth most common disease treated with antibiotics in adults in the USA and the most common diagnosis that received out-patient antibiotic prescriptions [2, 8]. For ARS, 84–87% of Canadian outpatients [9] and 94% of adults in primary care clinics in midwestern USA [10] received antibiotics. Considering the association between antibiotic consumption and microbial resistance [11, 12], there have been international efforts to reduce antibiotic prescriptions [13, 14].

Other managements for RS include herbal medicines. A review of 10 randomised controlled trials (RCTs) of herbal medicines for ARS or CRS found limited evidence of benefit [15]. A review that included some herbal medicines used in eastern Asia found symptom improvements in CRS [16]. In China and other countries, Chinese herbal medicines (CHMs) have been used for nasal disorders since ancient times. References to a disorder that may have been sinusitis, then called *bi yuan* (excessive turbid nasal discharge), appeared in the book *Huang Di Nei Jing* (Yellow Emperor's Classic of Medicine) which dates back to the Han dynasty (c. 206 CE—220 CE), and subsequently appear in multiple books until modern times [17, 18]. Currently, CHMs can be used as integrative therapies [19, 20]. One review of 32 RCTs of CHMs following surgery found benefits for adding CHM nasal irrigations to conventional therapies [21].

This systematic review aims to: assess the effects and safety of CHMs in the management of ARS and CRS; inform clinicians of the current state of the evidence; identify the best available evidence; and suggest directions for further research. The research question was whether CHMs administered orally and/or nasally improved scores on measures of RS symptoms, sinus imaging or measures of mucociliary clearance.

## Materials and methods

This review followed the PRISMA guidelines [22, 23] and the methods of the Cochrane Collaboration [24, 25]. The protocol for this systematic review was registered with PROSPERO (CRD42019119586).

### Selection criteria

Included studies were prospective RCTs with no limitations on language or publication type.

**Participants:** included adults and/or children who were diagnosed with acute or chronic sinusitis/rhinosinusitis based on guidelines [1, 2, 26–32]. Studies without clear diagnostic criteria, that included participants with non-RS conditions, only included participants with nasal polyps, or were of post-surgical recovery were excluded.

**Test interventions:** were CHMs used in eastern Asia (China, Korea, Japan) administered orally and/or nasally. Forms could include liquids, steam inhalations, sprays, granules, capsules or pills. Injections, purified compounds, and displacement techniques were excluded.

**Control interventions:** included placebos, no additional treatment, and conventional non-invasive treatments including pharmacotherapies (oral and/or nasal), nasal irrigations, and/or inhalations, and/or inhalations as in guidelines [1, 2, 26, 28–32]. Invasive procedures such as surgery were excluded. Non-invasive co-interventions were allowed when used in both groups.

**Outcome measures:** were Sino-Nasal Outcome Test (SNOT), visual analogue scales for total nasal symptoms (VAS-TNS) and/or individual symptoms (VAS-IS), Lund-Mackay computed tomography score (LM), Lund-Kennedy endoscopic score (LK), Mucociliary transport time (MTT), Mucociliary transport rate (MTR) and/or Short-Form 36 (SF-36®). Categorical scales such as effective rates, scales not used internationally, and measures developed by the authors were excluded.

**Settings:** included in-patients and out-patients. Post-surgical recovery was excluded.

**Information sources and search strategy.** Five English language and four Chinese language databases were searched from their respective inception dates, with no limits on years, until August 9th 2022. Additional sources included four clinical trial registries, Web of Science, ProQuest Central which were searched from their inception dates until August 12th 2022 with no limits; and we searched reference lists in retrieved papers. The information sources and search terms are listed in S1 Table.

**Data screening and extraction.** Search results were screened according to the selection criteria by JC, WML and BHM based on titles and abstracts. Full texts of possible inclusions were obtained for further screening by two reviewers. For included studies, the characteristics, funding sources and outcome data were extracted to predefined spreadsheets by JC and WML, checked by JC, BHM and QL independently, and analysed in Review Manager 5.4.1. Any issues were resolved by discussion between reviewers, with ALZ as final arbiter. Plant names were based on the Chinese pharmacopoeia [33].

## Risk of bias assessment

Risk of bias was assessed by two reviewers (JC, BHM) independently and mediated by a third (ALZ) using the Cochrane tool [24] for sequence generation (SG), allocation concealment (AC), blinding of participants (BPt), blinding of personnel (BPn), blinding of outcome assessment (BOA), incomplete outcome data (IOD), and selective outcome reporting (SOR). Reporting bias was assessed using Funnel plots and Egger's test when ten or more studies were available.

**Data analysis.** Analysis was conducted in Review Manager 5.4.1. Mean difference (MD) and risk ratio (RR) were assessed using and 95% confidence intervals (CI) with heterogeneity as $I^2$. Due to likely heterogeneity in study populations and methods, conservative random-effect models were used. Baseline scores were assessed between groups to determine baseline comparability. Planned sensitivity analyses explored any effects of baseline imbalance, study duration, use of same CHM, and use of same pharmacotherapy. Grading of Recommendations Assessment, Development and Evaluation (GRADE) was used to assess the certainty of the evidence [34, 35].

## Results

### Literature search results

Search results were downloaded to spreadsheets and combined. After removal of obvious duplications, 8,785 records were screened. Based on titles, abstracts and other information, 8,395 records were excluded, and 390 full text papers were obtained for further assessment against the inclusion and exclusion criteria. Thirty-four RCTs satisfied selection criteria

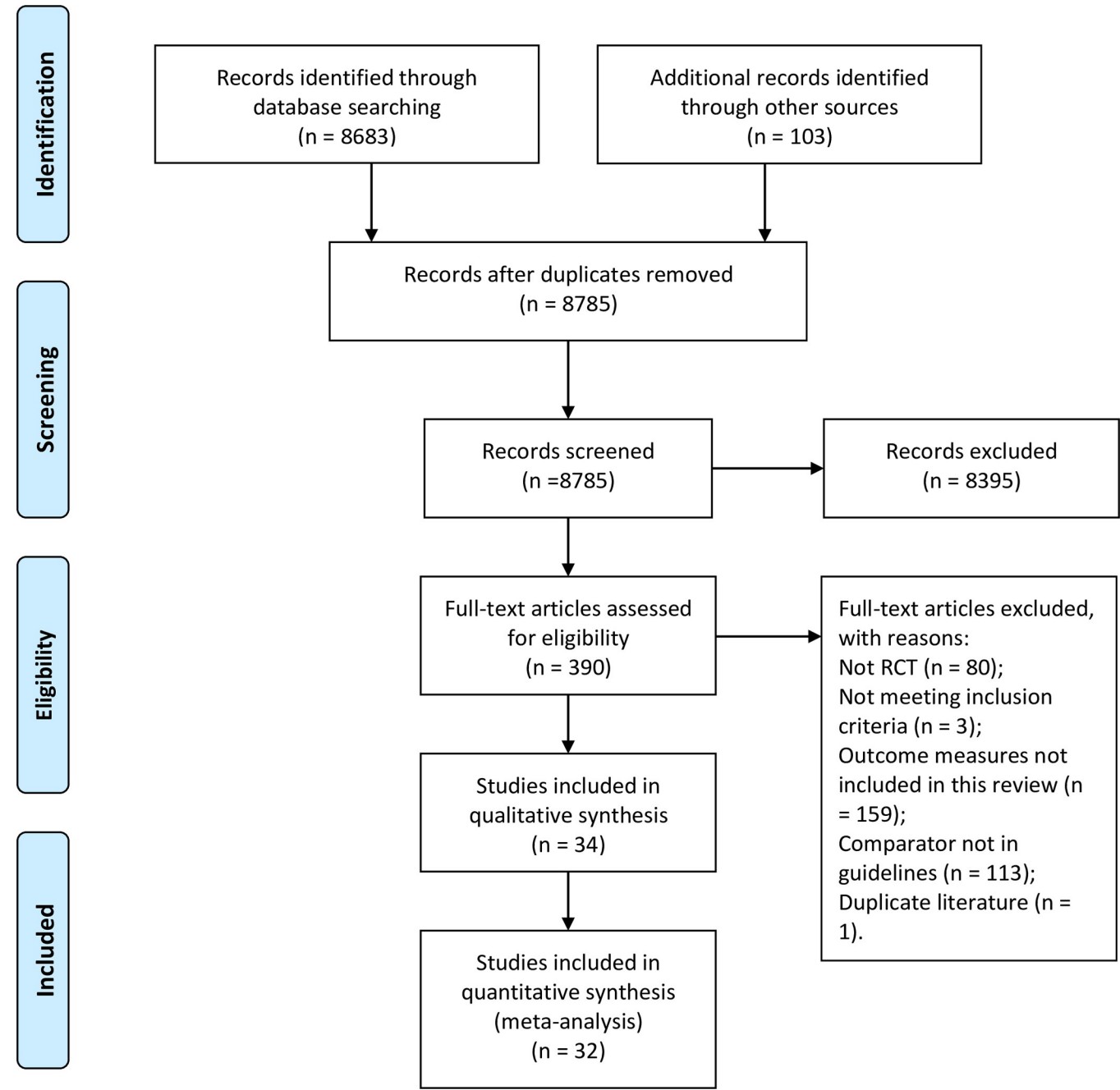

**Fig 1. Flow diagram of search and selection process for studies of CHM for RS.** Abbreviations: CHM: Chinese herbal medicine; RCT: randomized controlled trial; RS: rhinosinusitis.

(Fig 1). Two were written in English [36, 37] and 32 in Chinese. These enrolled 3,752 partici-pants aged six to 86 years. Four were of ARS [38–41]. Thirty were of CRS [36, 37, 42–69]. Four included children and/or adolescents [56, 57, 67, 69]. Treatment durations ranged from three days to 16 weeks. Four studies included three or more groups [38, 39, 53, 68]. A list of potential studies that were excluded, with reasons is included in S1 Table.

## Interventions

Five studies tested nasally-administered CHMs [42, 56–58, 66], two combined nasal plus oral CHMs [39, 63] and 25 used oral CHMs. In total, the 34 RCTs tested 24 different CHM formulae (Table 1). The most frequent was *Bi yuan shu kou fu ye/Bi yuan shu jiao nang* (BYSKFY/BYSJN) which have the same ingredients but different preparation forms (4 studies). Two formulae, *Bi yuan tong qiao ke li* (BYTQKL) and *Bi dou yan kou fu ye* (BDYKFY), were used in three studies each. The following four formulae were used in two studies each: (LHQWKL), *Long dan xie gan tang* (LDXGT), *Bi yan kang tang* (BYKT) and *Cang er zi san* (CEZS) including modified versions. Two studies used a CHM with the same name, *Bi yuan tang* [57, 59], but different ingredients (designated BYT1, BYT2). Pharmacotherapies were mainly macrolide antibiotics [70]. For details of test and control interventions, administration instructions, manufacturing, and sources of study funding see S1 Table.

Despite differences in names, many CHMs shared common ingredients (S1 Table). The most frequent ingredients were *Magnolia biondii* Pamp. (*xin yi*) n = 30, *Angelica dahurica* (Fisch. ex Hoffm.) Benth. et Hook. f (*bai zhi*) n = 29, *Xanthium sibiricum* Patr. (*cang er zi*) n = 26, and *Scutellaria baicalensis* Georgi (*huang qin*) n = 22.

## Risk of bias

Twenty studies were judged 'low' risk for SG (S1 Table). One that compared oral CHM with an identical placebo [43] was also judged 'low' risk for BPt but 'unclear' for AC, BPt, BPn, and BOA due to lack of clear descriptions. One that compared a CHM steam inhalation with a placebo inhalation [42] was judged 'low' risk for SG, AC, BPt, BPn, and BOA. Two studies of oral CHMs used 'double-dummy' designs [36, 44]. Both were judged 'low' risk for SG, AC, BPt, BPn, and BOA. The remaining studies were judged 'unclear' for AC and 'high' for blinding domains. Two were judged 'unclear' for IOD since there were >20% dropouts without reasons [36, 58]. One was judged high risk for SOR [64] since one of the outcomes mentioned in the methods was not mentioned in the results. All were judged 'unclear' for SOR, as study protocols were unavailable. It was not possible to assess potential publication bias since no comparison included ten or more studies.

## Comparisons

One study compared oral CHM with no treatment [38]; one compared CHM steam inhalation to inhalation of steam from distilled water [42]; one compared an oral CHM decoction with a placebo decoction [43], and one compared oral CHM granules with placebo granules [37]. Eight studies compared CHMs with active controls. Two used 'double-dummy' designs [36, 44] and six were open-label [38, 39, 45, 66, 68, 69]. In 23 studies the test groups combined CHMs with pharmacotherapies (PT) as integrative medicine (IM).

Outcomes were between-group scores at end of treatment (EoT) and/or end of follow-up (FU). To examine effect sizes, within-group changes (baseline versus EoT) were assessed for test and control groups.

## Chronic rhinosinusitis

Thirty studies reported one or more of the following outcomes.

**Sino-Nasal Outcome Test.** For oral LDXGT versus placebo [43], SNOT-20 reduced in both groups, with a greater reduction in the LDXGT group (MD -4.90 [-8.12, -1.68]), despite a baseline imbalance in favour of the placebo group (S1 Table). Another study reported SNOT-22 as median scores which the authors stated were significantly lower in the oral *Lian hua qing*

**Table 1. Characteristics of included studies of Chinese herbal medicines for rhinosinusitis by comparison.**

| Study name; Duration | Diagnosis; N. participants (T,C); Age | Intervention | | Outcome measures included in this review |
|---|---|---|---|---|
| | | Test group (T) | Control group (C) | |
| **CHM versus inactive controls**: 4 studies (4 test groups) | | | | |
| Lin L 2015a; 2wks | ARS; (34,28); 18–66 yrs | **Oral** *Lian hua qing wen ke li* (LHQWKL) | no treatment | SNOT-22 (in figure only) |
| Lin L 2020; 30 days | CRS; (70,70) (72,72 at BL); 18–75 yrs | **Oral** *Lian hua qing wen ke li* (LHQWKL) | placebo | SNOT-22, VAS-TNS (median scores, figures) |
| Yang L 2010; 3 days | CRS; (26,26); mean T 38.65 yrs / C 34.81 yrs | **Nasal** steam inhalation of unnamed CHM decoction | nasal placebo: steam inhalation of distilled water | VAS-IS; MTT |
| Zhou L 2013; 2wks | CRS; (30,30); 16–60 yrs | **Oral** *Long dan xie gan tang* (LDXGT) | oral placebo for LDXGT | SNOT-20; VAS-TNS; SF-36 |
| **CHM versus active controls (pharmacotherapies)**: 8 studies (10 test groups) | | | | |
| Jiang RS 2012; 8wks | CRS; (26,27); 18–86 yrs | **Oral** *Cang er zi san* (CEZS) mod. plus a placebo for erythromycin | oral erythromycin plus a placebo for the CHM | SNOT-20; LM |
| Li MJ 2014a & b; 2wks | ARS; (a. 32; b. 36; c. 29); 20–60 yrs | a. **Oral** *Cang er zi san* (CEZS) mod.; b. **Oral + nasal** *Cang er zi san* (CEZS) mod. | c. oral amoxicillin | VAS-blockage |
| Lin L 2015b & c; 2wks | ARS; (a. 34; b. 30; c. 31); 18–66 yrs | **Oral** *Lian hua qing wen ke li* (LHQWKL) | b. oral amoxicillin or c. budesonide nasal spray | SNOT-22 (in figure only) |
| Liu JB 2011; 12wks | CRS; (24,24); 18–54 yrs | **Oral** *Bi yuan shu jiao nang* (BYSJN) plus a placebo for clarithromycin | oral clarithromycin plus a placebo for the CHM | VAS-TNS; LM |
| Qiang JH 2011; 4wks | CRS; (a. 30; b. 30; c. 25); 11–71 yrs | a. **Oral** *Bi yuan he ji* (BYHJ) mod.[1] b. **Oral** *Bi yan pian* (BYP) | c. oral erythromycin | MTT |
| Wang J 2020; 2wks | CRS; (30,31); 21–56 yrs (32,32 at BL) | **Nasal** *Yu jiang pai du he ji* (YJPDHJ) | 0.9% saline | SNOT-22;[2] VAS-TNS; LK; LM, MTT (min) |
| Wu MM 2022; 4wks | CRS; (35,34); 15–68 yrs (36,36 at BL) | **Oral** *Tong qiao xiao ti ke li* (TQXTKL) | oral clarithromycin | VAS-TNS; LK; LM |
| Xiong J 2016; 4wks | CRS; (40,40); mean T 38.76 / C 37.59 yrs | **Oral** *Long dan tong qiao wan* (LDTQW) | oral clarithromycin plus fluticasone nasal | VAS-TNS; LK |
| **CHM plus pharmacotherapy (integrative medicine) versus the same pharmacotherapy (PT)**: 23 studies (24 test groups) | | | | |
| Cai CJ 2019; 4wks | CRS; (28,27); 18–75 yrs | **Oral** *Gan lu xiao du dan* (GLXDD) plus PT | mometasone furoate nasal spray | SNOT-20; LK, LM (no data) |
| Chen TT 2017; 12wks | CRS; (30,30); 18–65 yrs | **Oral** *Long dan xie gan tang* (LDXGT) plus PT | oral clarithromycin plus mometasone furoate nasal spray | SNOT-20; LM; LK; SF-36® |
| Chu XY 2017; 10d | CRS; (63,63); mean T 41.2 / C 41.7 | **Oral** *Bi yuan tong qiao ke li* (BYTQKL) plus PT | triamcinolone acetonide nasal spray | MTT |
| Dai RZ 2015; 15d | CRS; (58,58); 16–58 yrs | **Oral** *Bi yuan shu kou fu ye* (BYSKFY) plus PT | oral clarithromycin | LK |
| Deng QH 2016; 30d | CRS; (74,74); 18–58 yrs | **Oral** *Bi yuan tong qiao ke li* (BYTQKL) plus PT | triamcinolone acetonide nasal spray | MTT |
| Du JW 2016; 20d | CRS; (44,45); mean T 33.2 / C 32.25 yrs | **Oral** *Bi dou yan kou fu ye* (BDYKFY) plus PT | oral clarithromycin | LK |
| Guo L 2015; 4, wks, 12wks FU | CRS; (60,60); 12–65 yrs | **Oral + nasal** *Bi yan kang tang* (BYKT) plus PT | oral clarithromycin plus saline and dexamethasone nasal wash | SNOT-20; VAS-TNS; LK |
| Hong HY 2015; 4wks | CRS; (32,32); 18–60 yrs | **Oral** *Bi yuan tong qiao ke li* (BYTQKL) plus PT | oral clarithromycin | SNOT-20; VAS-IS; LK |
| Hu FL 2015; 12wks | CRS; (42,42); mean T 60.5 / C 60.2 yrs | **Oral** *Bi yuan shu kou fu ye* (BYSKFY) plus PT | oral clarithromycin | LM |
| Huang JY 2017; 10d | ARS; (95,95); mean T 36.6 / C 35.9 yrs | **Oral** *Bi dou yan kou fu ye* (BDYKFY) plus PT | oral cefuroxime tablets | VAS-TNS; LM; LK |

*(Continued)*

**Table 1.** (Continued)

| Study name; Duration | Diagnosis; N. participants (T,C); Age | Intervention | | Outcome measures included in this review |
|---|---|---|---|---|
| | | Test group (T) | Control group (C) | |
| Liao WT 2020; 8 wks | CRS; (60,60); 16–64 yrs | **Oral** *Bi yan kang tang* (BYKT) **plus PT** | oral clarithromycin plus gentamicin sulfate injection + dexamethasone sodium phosphate injection + normal saline nasal wash | MTR mm/min |
| Liu HL 2017 CHMa,b; 4wks | CRS; (30,30,30); mean T 43.96 / C 44.66 yrs | **Oral CHMa:** *Bi yuan shu wan* (BYSW) **plus PT;** **Oral CHMb:** *Tong qiao bi yan ke li* (TQBYKL) **plus PT** | oral eucalyptol limonene and pinene capsules | VAS-TNS |
| Liu Q 2015 12wks | CRS; (43,43); 18–60 yrs | **Oral** *Bi dou yan kou fu ye* (BDYKFY) **plus the same PT** | oral clarithromycin | VAS-TNS; LM; LK |
| Wang C 2014; 12wks | CRS; (48,48); 12–75 yrs | **Oral** *Bi yuan shu kou fu ye* (BYSKFY) **plus PT** | oral clarithromycin plus eucalyptol limonene and pinene capsules plus saline nasal spray | VAS-TNS; LM |
| Wang G 2013; 20 days, 12wks FU | CRS; (49,49); 7–13 yrs | **Nasal spray** *Ma yi bi yan pen wu ji* (MYBYPWJ) **plus PT** | oral cefixime or roxithromycin | VAS-IS; LK |
| Wang H 2009; 3wks | CRS; (27,28); 6–14 yrs | **Oral** *Bi yuan gu ben fang* (BYGBF) **plus PT** | oral cefadroxil plus nasal spray of chloramphenicol plus dexamethasone | MTR |
| Wang KQ 2016; 4wks | CRS; (84,83); mean T 8.02 / C 8.07 yrs | **Nasal wash** *Bi yuan tang* (BYT1) **plus PT** | oral cefaclor plus budesonide nasal spray | VAS-IS |
| Wang P 2015; 16wks | CRS; (342,256); 18–66 yrs | **Nasal drop,** *Xin zhi di bi ye* (XZDBY) **plus PT** | oral roxithromycin plus fluticasone nasal spray | SNOT-20; VAS-TNS |
| Zhang LY 2015; 3wks | CRS; (40,40); 26.7–75.5 yrs | **Oral** *Bi yuan tang* (BYT2) **plus PT** | oral clarithromycin plus triamcinolone acetonide nasal spray | VAS-TNS |
| Zhang XQ 2015; 30d | CRS; (45,45); 22–70 yrs | **Oral** *Tong bi tang* (TBT) **plus PT** | oral penicillin plus oral metronidazole plus nasal drops ephedrine | SNOT-20 subscales; VAS-TNS |
| Zhang YF 2015; 12wks, 24 wks FU | CRS; (69,65); 17–62 yrs | **Oral** *Xiang ju jiao nang* (XJJN) **plus PT** | oral clarithromycin plus eucalyptol limonene and pinene capsules plus nasal budesonide | LM; LK |
| Zhong MR 2020; 1wk | ARS; (56,56); 19–65 yrs | **Oral** *Huang qin hua shi tang* (HQHST) **plus PT** | oral cefuroxime tablets | SNOT-22; VAS-TNS; LK |
| Zhu XP 2017; 28d | CRS; (48,48); mean T 35.2 / C 35.32 yrs | **Oral** *Xing qiao tang* (XQT) **plus PT** | oral amoxicillin plus nasal budesonide plus nasal spray containing sea water | VAS-TNS |

Abbreviations: ARS: acute rhinosinusitis; C: control group; CHM: Chinese herbal medicine; CRS: chronic rhinosinusitis; d: days; fig: figure; FU: follow-up; LK: Lund-Kennedy Endoscopic score; LM: Lund-Mackay computed tomography (CT) score; M: mean; mod: modified; MTR: mucociliary transport rate; MTT: mucociliary transport time; N: number; NS: not specified; PT: pharmacotherapies; SF-36Ⓡ: 36-Item Short Form Survey; SNOT-20 (22): Sino-Nasal Outcome Test (SNOT)-20 (22); T: treatment group;; VAS-IS: Visual Analog Scale—scores for individual symptoms; VAS-TNS: Visual Analog Scale—scores for total nasal symptoms; wks: weeks; yrs: years.

Notes: 1) BYHJ was clearly the focus of the study, so we have focussed on this CHM; 2) SNOT-22 results were as separate nasal symptoms only.

*wen ke li* (LHQWKL) group than in the placebo control at EoT [37]. A double-dummy study of oral CEZS versus erythromycin [36] found reductions within both groups and no difference between groups (MD-0.61 [-1.63, 0.41]).

In the open-label IM studies, the pooled result for three studies of oral CHMs showed a significantly greater improvement (MD -3.55 [-4.89, -2.21] $I^2$ = 0%, n = 179) without heterogeneity (Fig 2). One study of nasal *Xin zhi di bi ye* (XZDBY) [58] reported a greater improvement in the IM group; and a study of nasal plus oral *Bi yan kang tang* (BYKT) [63] reported a similar result. One IM study of oral *Tong bi tang* (TBT) that reported sub-scale data only [60] found significantly greater improvements for each sub-scale (S1 Table).

**Visual analogue scales.** Twelve studies (13 test groups) reported VAS-TNS. There was a greater reduction for oral LDXGT versus placebo [43] (MD -1.40 [-1.53, -1.27]) (Fig 3). The study of oral LHQWKL reported a greater reduction in the CHM group compared to placebo based on median scores (S1 Table) [37]. In the double-dummy study of oral *Bi yuan shu jiao*

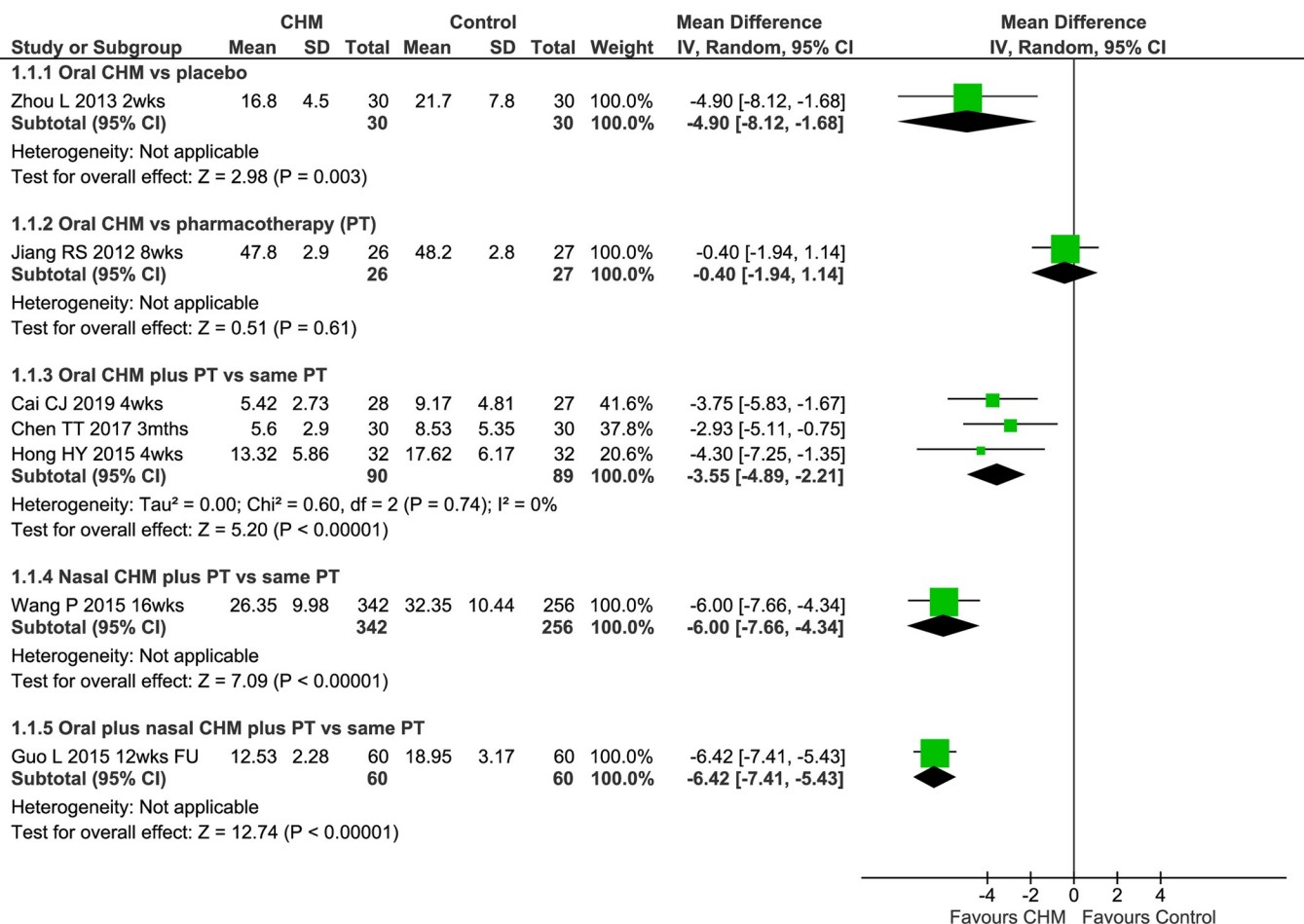

**Fig 2. Forest plot of CHM for CRS at end of treatment for SNOT-20 total score.** Abbreviations: CHM: Chinese herbal medicine; CRS: chronic rhinosinusitis; DB: double blind; FU: follow-up; SNOT-20: Sino-Nasal Outcome Test-20; PT: pharmacotherapy; vs: versus; wks: weeks.

*nang* (BYSJN) versus clarithromycin [44], both groups improved with no difference between groups. An open-label study of oral *Long dan tong qiao wan* (LDTQW) reported a similar result [45] as did an open-label study of oral *Tong qiao xiao ti ke li* (TQXTKL) [69]. The pooled result showed no significant difference (MD 0.03 [-1.06, 1.12] $I^2$ = 64%, n = 197) between the CHM and the antibiotic groups (all clarithromycin), but with substantial heterogeneity. The sensitivity analysis found a similar result.

In the six IM studies of oral CHMs, one included two test groups (Liu HL 2017a & b) [53]. In the pooled result of seven groups, there were greater reductions in VAS-TNS in the IM groups (MD -1.55 [-1.97, -1.13] $I^2$ = 91%, n = 538) with considerable heterogeneity. A sensitivity analysis of the two 12-week studies that used clarithromycin found a similar result (MD -1.52 [-2.46, -0.58] $I^2$ = 91%, n = 182) but the heterogeneity remained considerable. In one IM study of XZDBY nasal drops [58], both groups improved but there was no added benefit for the CHM (MD -0.13 [-0.40, 0.14]). For oral BYKT plus BYKT nasal wash [63] there was a benefit for adding the CHM (MD -0.73 [-1.11, -0.35]).

Four studies reported VAS-IS (S1 Table). The CHM steam inhalation showed greater reduction in nasal blockage compared with inactive inhalation [42]. All three IM studies reported greater reductions in nasal discharge [51, 56, 57] and there were significant improvements in other symptoms, but results were not poolable.

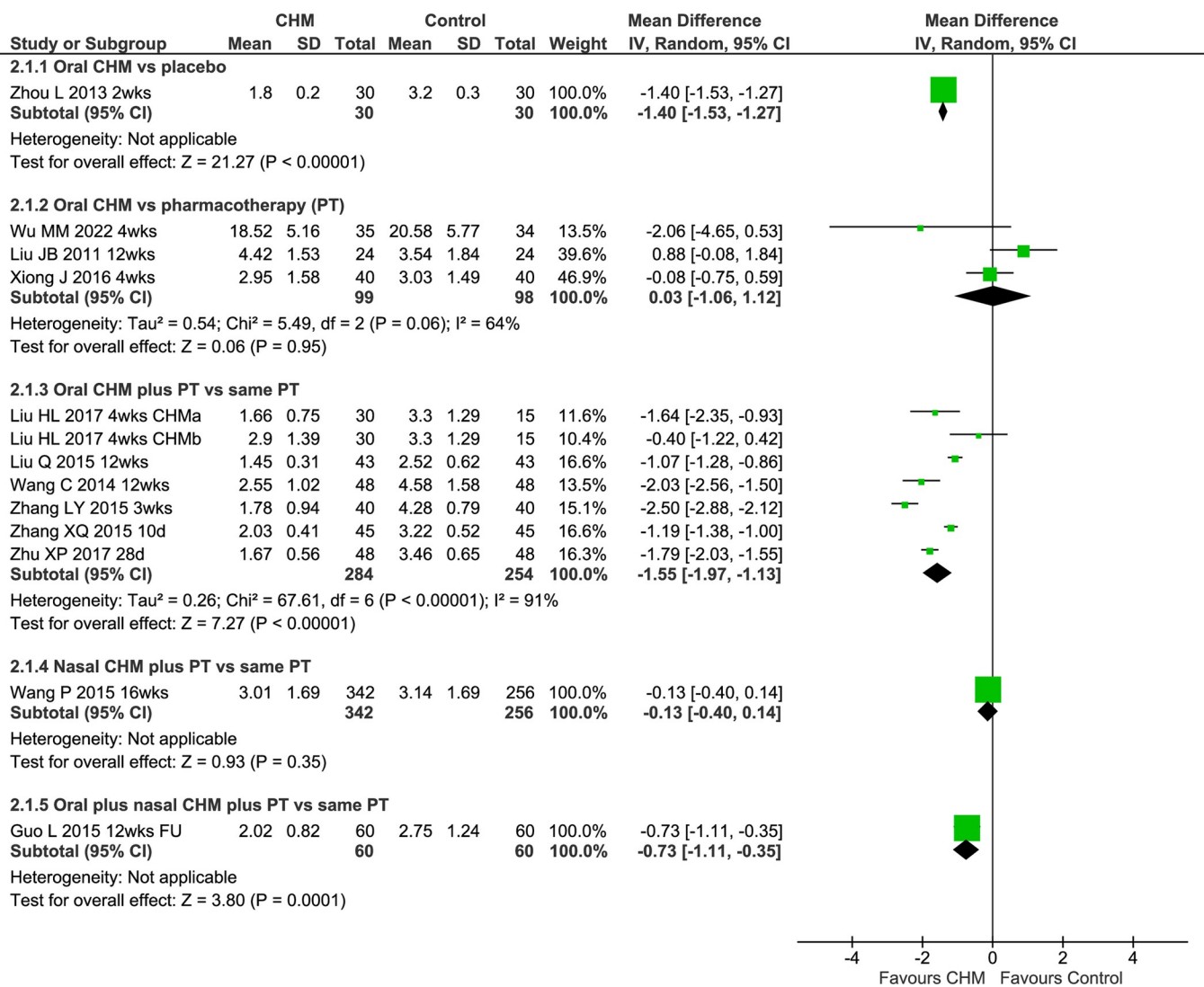

**Fig 3. Forest plot of CHM for CRS at end of treatment for VAS-TNS.** Abbreviations: CHM: Chinese herbal medicine; CRS: chronic rhinosinusitis; FU: follow-up; VAS: Visual analogue scale; TNS: total nasal symptoms; PT: pharmacotherapy; vs: versus; wks: weeks.

**Lund-Mackay computed tomography score.** Two double-dummy studies compared CHMs with pharmacotherapies (S1 Table). For oral CEZS both groups improved with no difference between groups [36]. In the other study [44], both groups improved with less improvement for oral BYSJN compared to clarithromycin but the BYSN group was significantly worse at baseline. An open-label study found no difference between groups for TBXTKL versus clarithromycin [69] (Fig 4). The pooled result showed no difference between groups (MD 0.31 [-0.78, 1.40] $I^2$ = 76%, n = 170) with considerable heterogeneity and the result was similar in the sensitivity analysis of blinded studies.

In the pooled result of four open-label studies, there were greater improvements in the IM groups at EoT (MD -1.51 [-1.98, -1.04] $I^2$ = 88%, n = 326) but the heterogeneity was considerable. In two 12-week studies of BYSKFY, the result was similar (MD -1.78 [-2.17, -1.39] $I^2$ = 73%, n = 180) with substantial heterogeneity. In another study that only provided data at 24 weeks follow-up [61], the IM group showed a significantly greater reduction in scores.

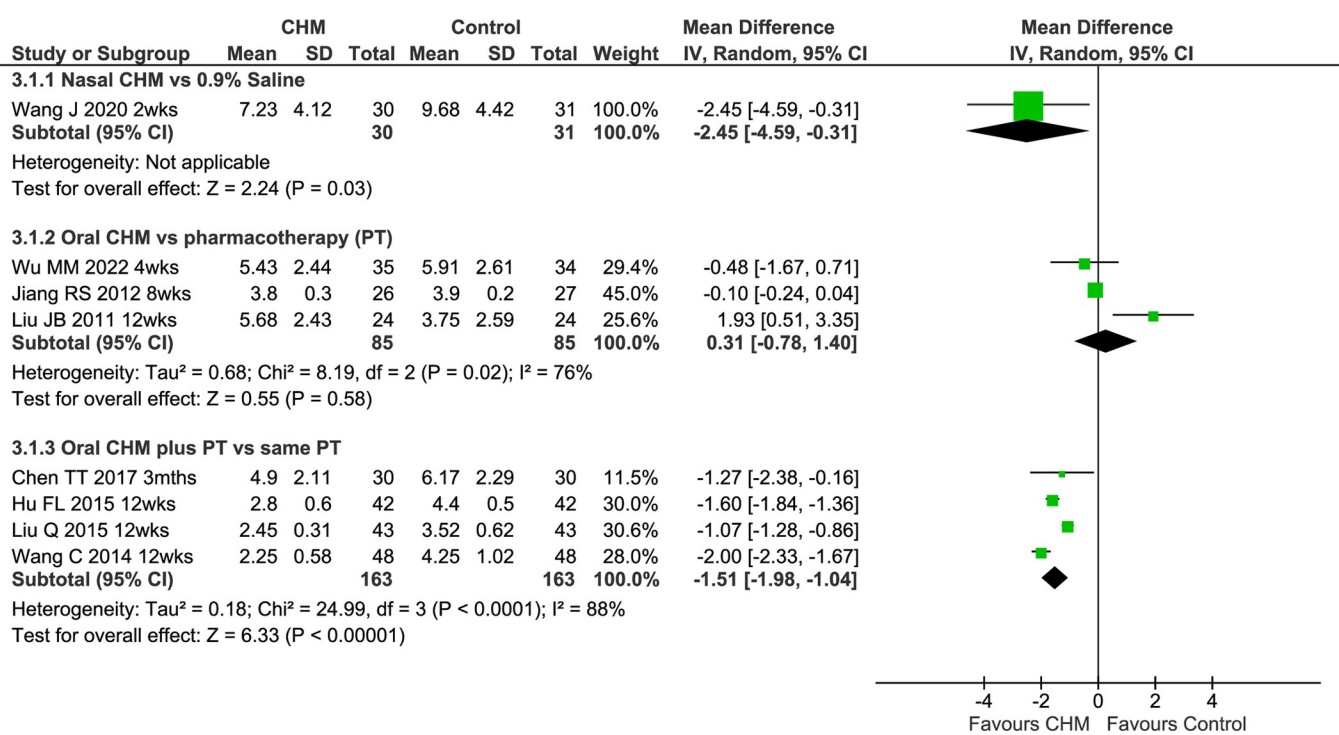

**Fig 4. Forest plot of CHM for CRS at end of treatment for LM.** Abbreviations: CHM: Chinese herbal medicine; CRS: chronic rhinosinusitis; DB: double blind; FU: follow-up; LM: Lund-Mackay computed tomography score; PT: pharmacotherapy; vs: versus; wks: weeks.

**Lund-Kennedy endoscopic score.** A study of oral LDTQW versus clarithromycin plus fluticasone nasal spray [45] and a study of oral TBXTKL versus clarithromycin [69] found greater improvements in the oral CHM groups (S1 Table). In the pooled result, the CHM groups showed a greater improvement in LK scores (MD -0.71 [-1.07, -0.35] $I^2$ = 0%, n = 149) without heterogeneity (Fig 5).

The pooled result of seven IM studies showed greater reductions in LK scores when oral CHMs were added to pharmacotherapies (MD -1.06 [-1.46, -0.65] $I^2$ = 85%, n = 604) with considerable heterogeneity. A sensitivity analysis of four studies with clarithromycin as controls found a similar result (MD -1.53 [-2.05, -1.02] $I^2$ = 81%). The pooled result for three 12-week studies showed a smaller but significant benefit (MD -0.72 [-1.29, -0.14] $I^2$ = 84%). The pooled result for two studies of BDYKFY also showed a significant reduction in LK scores in the IM groups (MD -1.73 [-3.08, -0.37] $I^2$ = 90%). However, the heterogeneity remained considerable in each of the three sensitivity analyses (S1 Table). For *Ma yi bi yan pen wu ji* (MYBYPWJ) nasal spray plus oral cefixime or roxithromycin [56] there was a greater improvement in the IM group. An IM study of oral BYKT plus BYKT nasal wash [63] also showed significantly reduced LK scores.

**Mucociliary transport time.** Steam inhalation of an herbal decoction compared to steam alone [42] showed greater reduction in the test group (S1 Table). In a three-group comparison between two CHMs and erythromycin [68], all groups improved with no significant differences between groups. Two IM studies of oral BYTQKL [47, 49] (Fig 6) showed a greater reduction in MTT in the IM groups (MD -224.90 [-308.68, -141.11] seconds $I^2$ = 0%, n = 274) without heterogeneity.

**Mucociliary transport rate.** In a study of oral *Bi yuan gu ben fang* (BYGBF) versus cefadroxil [67] there were improvements in both groups with a greater improvement in the

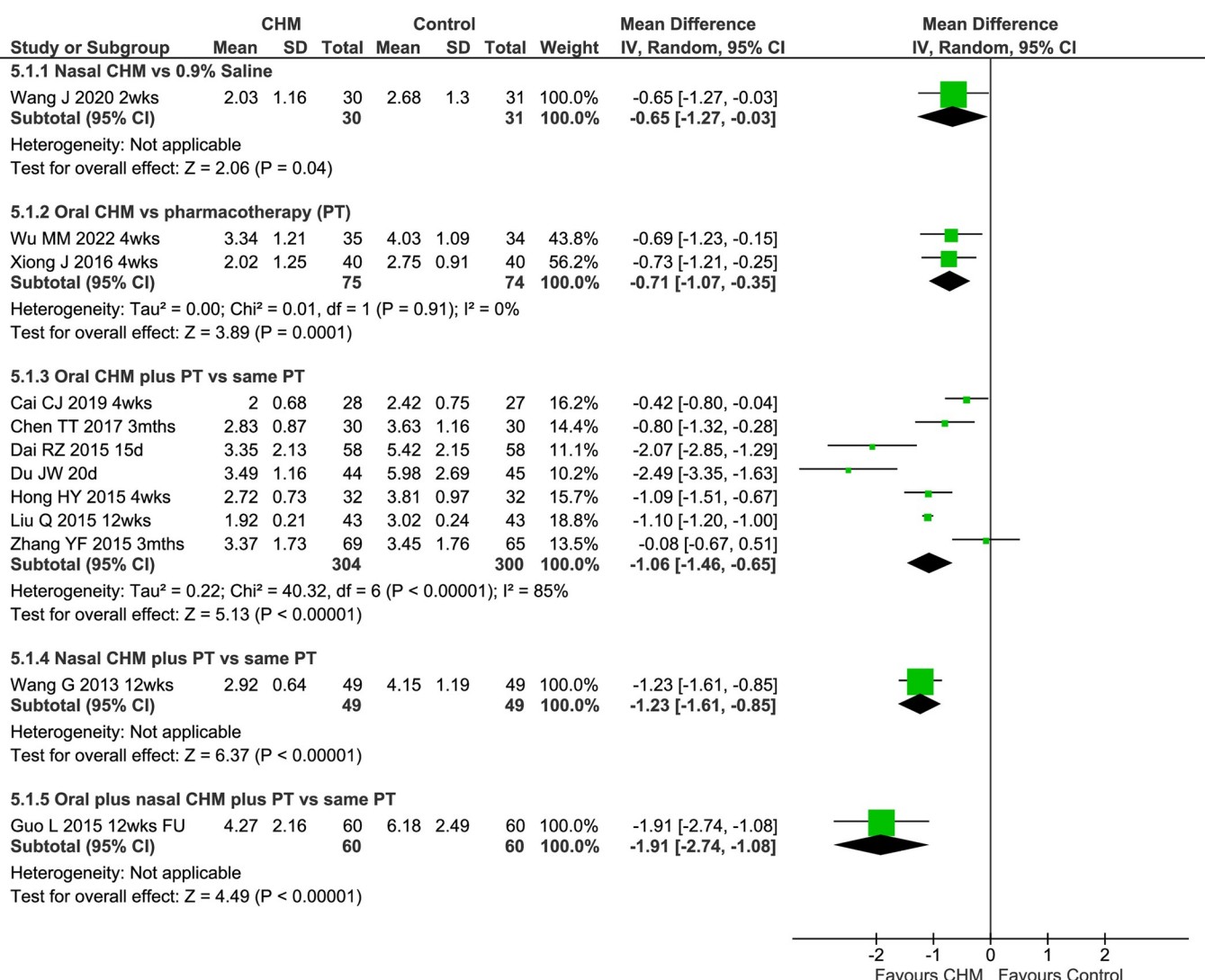

**Fig 5. Forest plot of CHM for CRS at end of treatment for LK.** Abbreviations: CHM: Chinese herbal medicine; CRS: chronic rhinosinusitis; FU: follow-up; LK: Lund-Kennedy endoscopic score; PT: pharmacotherapy; vs: versus; wks: weeks.

BYGBF group after three weeks (MD 1.11 [0.30, 1.92] mm/min) (S1 Table). In an IM study, both groups improved with greater improvements in the groups that also received oral BYKT (MD 1.43 [0.96, 1.90] mm/min) [65].

**Quality of life.** The total score on SF-36Ⓡ was reported for oral LDXGT versus placebo [43]. However, there was a large and significant difference between groups at baseline, so the EoT data was confounded. One IM study [46] of oral LDXGT reported data for eight SF-36Ⓡ subscales but not total score. The authors reported six subscales improved but our analyses found greater improvements in the IM group on four subscales (S1 Table).

## Acute rhinosinusitis

Two studies reported SNOT-22 (Table 2) but one was in figures only [38]. This study reported improvements in the oral LHQWKL, amoxicillin capsules, and budesonide spray groups but no differences between groups. The other study [41] reported a significantly greater reduction

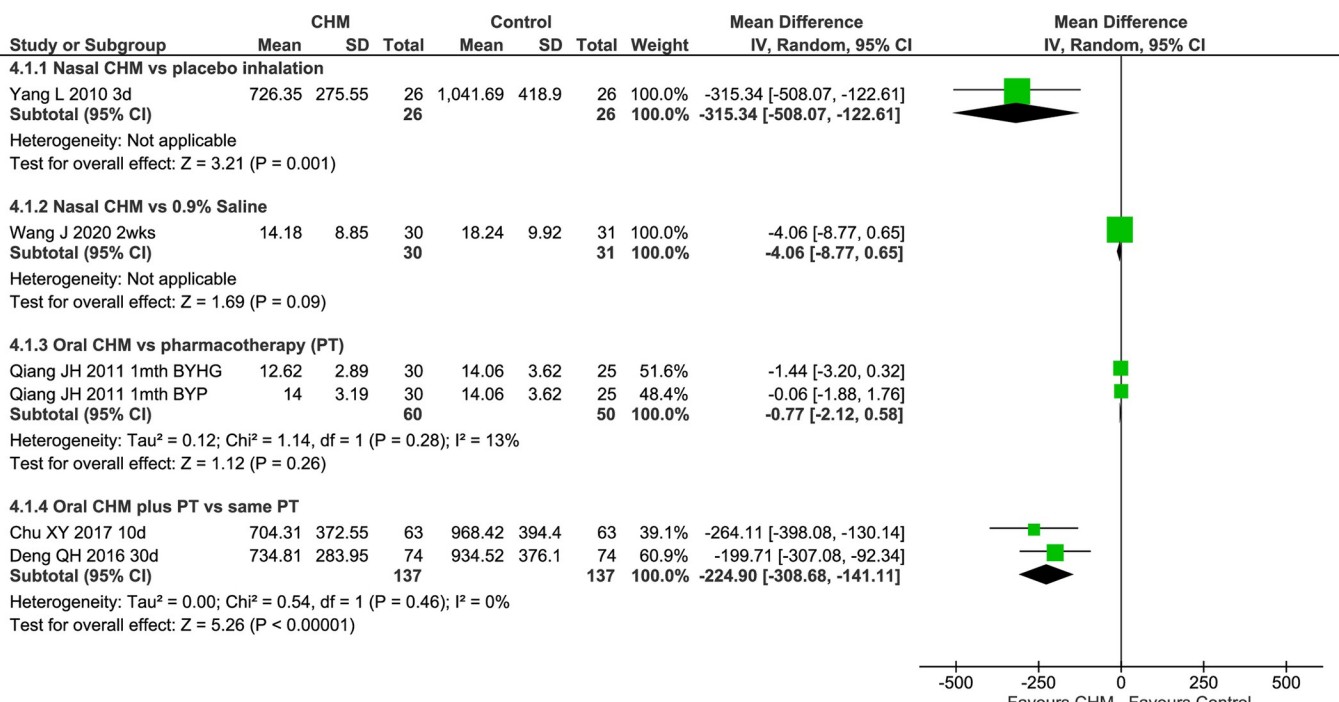

**Fig 6. Forest plot of CHM for CRS at end of treatment for MTT.** Abbreviations: CHM: Chinese herbal medicine; CRS: chronic rhinosinusitis; d: days; mth: month; MTT: Mucociliary transport time; PT: pharmacotherapy; vs: versus; wks: weeks.

in SNOT-22 in the group that combined oral *Huang qin hua shi tang* (HQHST) with oral cefuroxime tablets. There were similar results for VAS-TNS and LK.

In one three-group study [39] a decoction of CEZS was orally administered in one test group (Li MJ 2014a), while the other test group (Li MJ 2014b) used the hot decoction as a steam inhalation before drinking. VAS-IS (nasal obstruction) scores improved in all groups with no differences between groups. An RCT of oral BDYKFY plus cefuroxime [40] found VAS-TNS improved within both groups with a greater reduction in the integrative group. There were similar results for LM and LK.

The pooled VAS-TNS result for two IM studies showed a greater reduction in the groups that also received oral CHMs (MD -1.15 [-1.42, -0.88] $I^2$ = 81%). The pooled result for the two studies that reported LK was similar (MD -1.13 [-2.05, -0.21] $I^2$ = 98%) but the heterogeneity was considerable for each outcome.

## Adverse events

For CRS, in 16 studies there was no mention of AEs and nine reported there were no AEs in either group (S1 Table). Five studies reported specific AEs, but most were minor, and none led to withdrawals or dropouts. Although there were more AEs in the CHM groups (17 vs 7), there was no significant difference between groups (RR 2.06 [0.90, 4.69] $I^2$ = 0%, n = 514). Only one of the four studies of ARS mentioned AEs. This found no significant difference between groups. Overall, the AE data were insufficient for a complete safety analysis.

## GRADE assessments

GRADE assessments were conducted for orally administered CHMs for chronic RS. There were too few studies of nasal CHMs or of acute RS to warrant GRADE. Assessments were

**Table 2. Meta-analysis results for acute rhinosinusitis at end of treatment and changes within treatment and control groups.**

| Outcome measures | N studies (N participants at EoT: T, C) Duration [study name] | Test group (T) | Control group (C) | T vs C at EoT; MD [95% CI] | T group change (baseline vs EoT); MD [95% CI] | C group change (baseline vs EoT); MD [95% CI] |
|---|---|---|---|---|---|---|
| SNOT-22 | 1 (34,28) 2wks [Lin L 2015a] | Lian hua qing wen granules (LHQWKL) (oral) | no treatment | Reduction[1] | Reduction[1] | NA |
| | 1 (34,30) 2wks [Lin L 2015a] | | amoxicillin capsules | Reduction[1] | Reduction[1] | NA |
| | 1 (34,31) 2wks [Lin L 2015a] | | budesonide nasal spray | Reduction[1] | Reduction[1] | NA |
| | 1 (56,56) 1wk [Zhong MR 2020] | Huang qin hua shi tang (HQHST) (oral) IM | oral cefuroxime tablets | -4.44 [-5.66, -3.22]* No BI | -24.29 [-26.24, -22.34]* | -21.70 [-24.33, -19.07]* |
| VAS-nasal blockage[2] | 1 (32,29) 2wks [Li MJ 2014a] | Cang er zi san (CEZS), modified (oral) | amoxicillin | -0.61 [-1.63, 0.41] | 4.27 [-5.07, -3.47]* | 4.15 [-5.24, -3.06]* |
| | 1 (36,29) 2wks [Li MJ 2014b] | Cang er zi san (CEZS), modified (oral plus nasal) | amoxicillin | -0.94 [-1.92, 0.04] | -4.93 [-5.84, -4.02]* | -4.15 [-5.24, -3.06]* |
| VAS-TNS[2] | 1 (95,95) 10d [Huang JY 2017] | Bi dou yan kou fu ye (BDYKFY) (oral) IM | oral cefuroxime tablets | -1.30 [-1.50, -1.10]* | -5.49 [-5.82, -5.16]* | -4.26 [-4.59, -3.93]* |
| | 1 (56,56) 1wk [Zhong MR 2020] | Huang qin hua shi tang (HQHST) (oral) IM | oral cefuroxime tablets | -1.02 [-1.15, -0.89]* | -5.14 [-5.63, -4.65]* | -4.19 [-4.65, -3.73]* |
| VAS-TNS Pool | 2 (151, 151) 1 wk– 10 d | CHM (oral) IM | oral cefuroxime tablets | -1.15 [-1.42, -0.88]* 81% | -5.36 [-5.69, -5.03]* 27% | -4.24 [-4.50, -3.97]* 0% |
| LM[2] | 1 (95,95) 10d [Huang JY 2017] | Bi dou yan kou fu ye (BDYKFY) (oral) IM | oral cefuroxime tablets | -1.30 [-1.50, -1.10]* | -5.49 [-5.82, -5.16]* | -4.26 [-4.59, -3.93]* |
| LK[2] | 1 (95,95) 10d [Huang JY 2017] | Bi dou yan kou fu ye (BDYKFY) (oral) IM | oral cefuroxime tablets | -0.66 [-0.84, -0.48]* | -5.63 [-5.86, -5.40]* | -4.89 [-5.14, -4.64]* |
| | 1 (56,56) 1wk [Zhong MR 2020] | Huang qin hua shi tang (HQHST) (oral) IM | oral cefuroxime tablets | -1.60 [-1.74, -1.46]* no BI | -5.55 [-5.90, -5.20]* | -4.12 [-4.46, -3.78]* |
| LK Pool | 2 (151, 151) 1 wk– 10 d | CHM (oral) IM | oral cefuroxime tablets | -1.13 [-2.05, -0.21]* 98% | -5.61 [-5.80, -5.42]* 0% | -4.51 [-5.27, -3.76]* 92% |

* significant difference.

Abbreviations: C: control group; CI: confidence interval; d: days; EoT: end of treatment; IM: integrative medicine; LK: Lund-Kennedy Endoscopic score; LM: Lund-Mackay computed tomography (CT) score; MD: mean difference; N: number; NA: not applicable; SNOT-22: Sino-Nasal Outcome Test (SNOT)-22; T: treatment group; VAS: Visual Analog Scale scores for total nasal symptom score; VAS-TNS: Visual Analog Scale scores for total nasal symptom score; vs: versus; wks: weeks.

Notes: 1) based on author report and figures for summed scores; 2) baseline scores showed no significant difference between groups for each study in this comparison.

based on results for the clinically relevant outcome measures SNOT, VAS-TNS, LM and LK at the end of treatment for total pools (if available). Measures of mucociliary clearance were excluded. The certainty of evidence was downgraded for risk of bias (mainly blinding), significant heterogeneity ($I^2 \geq 50\%$, $p < 0.05$), small sample size (< 400 participants) and other issues. The comparisons were:

- Oral CHM versus placebo.

- Oral CHM versus pharmacotherapy (double-dummy studies).

- Oral CHM versus pharmacotherapy (all studies).

- Oral CHM plus pharmacotherapy versus pharmacotherapy (all studies).

## GRADE for oral Chinese herbal medicine versus placebo

For Oral CHM versus placebo only one RCT was available [43]. This used the formula LDXGT for two weeks versus a placebo decoction (Table 3). Assessable data were available for

**Table 3. GRADE for oral Chinese herbal medicine versus placebo for chronic rhinosinusitis.**

| Outcome (Duration) Study Name | Effect Size MD (95% CI) Heterogeneity N. Studies (N. Participants) | Certainty of Evidence GRADE |
|---|---|---|
| SNOT-20 at EoT (2 wks) [Zhou L 2013] | **4.90 pts lower** with CHM* (8.12 pts lower, to 1.68 pts lower) NA 1 study (60 participants) | ⊕⊕◯◯ **LOW** [1,2] |
| VAS-TNS at EoT (2 wks) [Zhou L 2013] | **1.40 cm lower** with CHM* (1.53 cm lower, to 1.27 cm lower) NA 1 study (60 participants) | ⊕⊕◯◯ **LOW** [1,2] |

*Statistically significant result, random effect model.

Abbreviations: CHM: Chinese herbal medicine; CI: confidence interval; cm: centimetres; EoT: end of treatment; GRADE: Grading of Recommendations Assessment, Development and Evaluation; MD: mean difference; N: number; NA: not applicable, pts: points; SNOT: Sino-Nasal Outcome Test; TNS: total nasal symptoms; VAS: visual analogue scale; wks: weeks.

Notes: 1. Baseline imbalance; 2. Small sample size.

two outcomes on which lower scores indicate an improvement. For SNOT-20, there was a mean reduction of 4.9 points in the CHM group compared to placebo which was a significant difference. For VAS-TNS the CHM group was 1.4 cm lower which was also a significant difference. The GRADE assessments for each outcome were downgraded by one grade for the presence of baseline imbalances. These were assessed as unlikely to have changed the direction of the effects or the significance tests, but they would have affected the magnitude of the effect size estimates. They were further downgraded due to the small sample size (n = 60), so the overall certainty of the evidence was judged as Low.

## GRADE for oral Chinese herbal medicine versus pharmacotherapy: Double-dummy studies

Two RCT employed placebo controls for both the CHM and the pharmacotherapy to enable double blinding [36, 44]. GRADE was conducted for three outcome measures (Table 4). For SNOT-20 scores, one double dummy study (n = 53) compared CEZS with erythromycin for eight weeks [36]. It found no significant difference between the two groups. The GRADE assessment was downgraded for small sample size (n = 53) and for the large number of dropouts (n = 30) since this may have affected the result which was per-protocol. So, the certainty of the evidence was judged to be Low. A different study [44] reported on VAS-TNS. This compared BYSJN with clarithromycin for 12 weeks. Again, there was no significant difference between the CHM and antibiotic groups. The GRADE assessment was downgraded by one grade for small sample size (n = 48). Therefore, the certainty of the evidence was judged as Moderate. Both studies reported results for LM scores. The pooled result (n = 101) showed no significant difference between the CHM and antibiotic groups after 8–12 weeks of treatment. However, one of the studies [44] found a greater improvement in the clarithromycin group leading to significant heterogeneity in the pooled result ($I^2$ = 87%). Therefore, the GRADE assessment was downgraded for both inconsistency and small sample size to Low.

## GRADE for oral Chinese herbal medicine versus pharmacotherapy: All studies

In total, four RCTs compared an oral CHM with a pharmacotherapy [36, 44, 45, 69] for chronic RS (Table 5). The GRADE assessment for SNOT-20 only included one study and was

**Table 4. GRADE for oral Chinese herbal medicine versus pharmacotherapy for chronic rhinosinusitis: Double-dummy studies.**

| Outcome (Duration) [Study Name] | Effect Size MD (95% CI) Heterogeneity N. Studies (N. Participants) | Certainty of Evidence GRADE |
|---|---|---|
| SNOT-20 at EoT (8 wks) [Jiang RS 2012] | **0.40 pts lower** with CHM (1.94 pts lower to 1.14 pts higher) NA 1 study (53 participants) | ⊕⊕◯◯ **LOW** [1,2] |
| VAS-TNS at EoT (12 wks) [Liu JB 2011] | **0.88 cm higher** with CHM (0.08 cm lower to 1.84 cm higher) NA 1 study (48 participants) | ⊕⊕⊕◯ **MODERATE** [1] |
| LM at EoT (8–12 wks) [Liu JB 2011, Jiang RS 2012] | **0.79 pts cm higher** with CHMs (1.19 pts lower to 2.76 pts higher) 87%* 2 studies (101 participants) | ⊕⊕◯◯ **LOW** [1,3] |

*Statistically significant result, random effect model.

Abbreviations: CHM: Chinese herbal medicine; CI: confidence interval; cm: centimetres; EoT: end of treatment; GRADE: Grading of Recommendations Assessment, Development and Evaluation; LM: Lund-Mackay computed tomography score; MD: mean difference; N: number; NA: not applicable, PT: pharmacotherapy; pts: points; SNOT: Sino-Nasal Outcome Test; TNS: total nasal symptoms; VAS: visual analogue scale; wks: weeks.

Notes: 1. Small sample size; 2. Large number of dropouts in Jiang RS 2012, over 20% dropouts (n = 30), no reasons given, no intent to treat analysis, but completers were balanced between groups; 3. Significant heterogeneity in pooled result.

**Table 5. GRADE for oral Chinese herbal medicine versus pharmacotherapy for chronic rhinosinusitis: All studies.**

| Outcome (Duration) [Study Name] | Effect Size MD (95% CI) Heterogeneity N. Studies (N. Participants) | Certainty of Evidence, GRADE |
|---|---|---|
| SNOT-20 at EoT (8 wks) [Jiang RS 2012] | **0.40 pts lower** with CHM (1.94 pts lower to 1.14 pts higher) NA 1 study (53 participants) | ⊕⊕◯◯ **LOW** [1,2] |
| VAS-TNS at EoT (4–12 wks) [Wu MM 2022, Liu JB 2011, Xiong J 2016] | **0.03 pts higher** with CHM (1.06 pts lower to 1.12 pts higher)] 64% 3 studies (197 participants) | ⊕⊕◯◯ **LOW** [1,3] |
| **LM** at EoT (4–12 wks) [Jiang RS 2012, Liu JB 2011, Wu MM 2022] | **0.31 pts higher** with CHM (0.78 pts lower to 1.40 pts higher) 76%* 3 studies (170 participants) | ⊕◯◯◯ **VERY LOW** [1,3,4] |
| **LK** at EoT (4 wks) [Xiong J 2016, Wu MM 2022] | **0.71 pts lower** with CHM* (1.07 pts lower to 0.35 pts lower) 0% 2 studies (149 participants) | ⊕⊕◯◯ **LOW** [1,3] |

*Statistically significant result, random effect model.

Abbreviations: CHM: Chinese herbal medicine; CI: confidence interval; cm: centimetres; EoT: end of treatment; GRADE: Grading of Recommendations Assessment, Development and Evaluation; LK: Lund-Kennedy endoscopic score; LM: Lund-Mackay computed tomography score; MD: mean difference; min: minutes; mm: millimetre; N: number; PT: pharmacotherapy; pts: points; SNOT: Sino-Nasal Outcome Test; TNS: total nasal symptoms; VAS: visual analogue scale; wks: weeks.

Notes: 1. Small sample size; 2. Large number of dropouts in Jiang RS 2012, over 20% dropouts (n = 30), no reasons given, no intent to treat analysis, but completers were balanced between groups; 3. Lack of blinding in at least one study; 4. Significant heterogeneity in pooled result.

the same as in Table 4. For VAS-TNS, three studies [44, 45, 69] compared oral CHMs with pharmacotherapies and found there was no significant difference between groups after four to 12 weeks. The heterogeneity was 64% but this was not statistically significant (p = 0.06). The GRADE assessment was downgraded by two levels for lack of blinding in two of the studies and small sample size (n = 197) to Low. Three studies reported LM [36, 44, 69] and the pooled result showed no significant difference between groups with significant heterogeneity. The GRADE assessment was rated down three levels for lack of blinding in one of the studies, small sample size (n = 170) and significant heterogeneity to Very Low. For LK, the pooled result of two studies [45, 69] showed a significantly greater reduction in the CHM group. The GRADE assessment of the certainty of this evidence was rated down for lack of blinding and small sample size (n = 149) to Low.

## GRADE for oral Chinese herbal medicine plus pharmacotherapy versus pharmacotherapy: All studies

The pooled results of three IM studies [46, 51, 64] found a significantly greater reduction in SNOT-20 scores in the IM groups without heterogeneity (Table 6). The GRADE of the evidence was rated down for lack of blinding and small sample size (n = 179) to Low. For VAS-TNS there were six studies of three to 12 weeks duration [53–55, 59, 60, 62] and since one study tested two different CHMs, there were seven groups (n = 538). There was a significantly greater reduction in symptoms the IM groups but there was significant heterogeneity. Therefore, the GRADE was rated down for lack of blinding and heterogeneity to Low. Four

**Table 6. GRADE for oral Chinese herbal medicine plus pharmacotherapy versus pharmacotherapy for chronic rhinosinusitis: All studies.**

| Outcome (Duration) [Study Name] | Effect Size MD (95% CI) Heterogeneity N. Studies (N. Participants) | Certainty of Evidence, GRADE |
|---|---|---|
| SNOT-20 at EoT (4–12 wks) [Chen TT 2017, Hong HY 2015, Cai CJ 2019] | **3.55 pts lower** with CHMs* (4.89 pts lower to 2.21 pts lower) 0% 3 studies (179 participants) | ⊕⊕◯◯ **LOW** [1,2] |
| VAS-TNS at EoT (3–12 wks) [Liu HL 2017 CHM1, Liu HL 2017 CHM2, Liu Q 2015, Wang C 2014, Zhang LY 2015, Zhang XQ 2015, Zhu XP 2017] | **1.55 cm lower** with CHMs* (1.97 cm lower to 1.13 cm lower) 91%* 6 studies, 7 groups (538 participants) | ⊕⊕◯◯ **LOW** [2,3] |
| **LM** at EoT (12 wks) [Chen TT 2017, Hu FL 2015, Liu Q 2015, Wang C 2014] | **1.51 pts lower** with CHMs* (1.98 pts lower to 1.04 pts lower) 88%* 4 studies (326 participants) | ⊕◯◯◯ **VERY LOW** [1,2,3] |
| **LK** at EoT (15 days-12 wks) [Cai CJ 2019, Chen TT 2017, Dai RZ 2015, Du JW 2016. Hong HY 2015, Liu Q 2015, Zhang YF 2015] | **1.06 pts lower** with CHMs* (-1.46 pts lower to -0.65 pts lower) 85%* 7 studies (604 participants) | ⊕⊕◯◯ **LOW** [2,3] |

*Statistically significant result, random effect model.

Abbreviations: CHM: Chinese herbal medicine; CI: confidence interval; cm: centimetres; EoT: end of treatment; GRADE: Grading of Recommendations Assessment, Development and Evaluation; LK: Lund-Kennedy endoscopic score; LM: Lund-Mackay computed tomography score; MD: mean difference; min: minutes; mm: millimetre; N: number; PT: pharmacotherapy; pts: points; SNOT: Sino-Nasal Outcome Test; TNS: total nasal symptoms; VAS: visual analogue scale; wks: weeks.

Notes: 1. Small sample size; 2. Lack of blinding; 3. Significant heterogeneity in pooled result.

studies reported LM at 12 weeks and the pooled result showed a significantly greater reduction in LM scores in the IM group with significant heterogeneity. The GRADE assessment was rated down for lack of blinding, small sample size (n = 326) and heterogeneity to Very Low. For LK, seven studies with durations ranging from 15 days to 12 weeks were included [46, 48, 50, 51, 54, 61, 64]. In the pooled result (n = 604) there was a significantly greater reduction in LK scores in the IM group. Lack of blinding and significant heterogeneity led to a GRADE assessment of Low.

## Discussion

For CRS, CHMs compared with placebo reported significant reductions in SNOT-20 [43], SNOT-22 [37], VAS-TNS [37, 43], and MTT [42]. A double-dummy study of CEZS versus erythromycin (n = 53) found that both interventions produced similar reductions in SNOT-20 and LM [36]. A double-dummy study (n = 48) of oral BYSJN versus clarithromycin [44] reported improvements in VAS-TNS within both groups with no difference between groups. The improvement in LM was less in the BYSJN group but this may have been due to the CHM group being worse at baseline. These blinded studies suggest possible effectiveness of these CHMs, but the limitations are small sample sizes and lack of replication. GRADE assessments for the placebo-controlled study judged the certainty of the evidence as Low, while the double-dummy studies were judged as Moderate to Low certainty. When all comparisons with pharmacotherapies were combined, the GRADE assessments were Low to Very Low certainty.

The largest result pools for oral CHMs were IM studies that reported LK (n = 604), VAS-TNS (n = 538) and LM (n = 326). These pooled results suggested additional improvements when oral CHMs were combined with pharmacotherapies, but all were open label, the effect sizes were variable, and the GRADE assessments were Low to Very Low certainty.

For nasal CHMs, one placebo-controlled study (n = 52) found a CHM steam inhalation improved VAS-IS (nasal blockage) and MTT [42]. One large (n = 598) open-label IM study of XZDBY nasal drops reported additional improvements on SNOT-20 [58]. In children with CRS, open-label IM studies of MYBYPWJ nasal spray (n = 98) and BYT nasal wash (n = 167) reported improvements in VAS-IS for nasal blockage and discharge [56, 57] but the data could not be pooled. Overall, there was inadequate data for any strong conclusions regarding nasal CHMs.

In ARS, the pooled results of two studies found the addition of oral CHMs to pharmacotherapies improved VAS-TNS and LK after ten days to two weeks of treatment but the studies were not blinded [40, 41], precluding any strong conclusions regarding the effectiveness of CHM in ARS.

The CHMs used in three or four studies each (BDYKFY, BYSKFY/BYSJN, and BYTQKL) were all commercial products that may not be available outside China. For CRS, pooled results for two IM studies showed significant benefits for BYSKFY on LM (2 studies, n = 180) [52, 55], for BDYKFY on LK (2 studies, n = 178) [50, 54], and BYTQKL on MTT (2 studies, n = 274) [47, 49]. In ARS, BDYKFY showed benefits for VAS-TNS, LM and LK but this was based on a single study [40].

The widely-available traditional formula, LDXGT, was tested in one placebo-controlled study [43] and one open label IM study [46] of CRS in adults. Both showed improvements on SNOT-20 and VAS-TNS, but results were not poolable. Another well-known traditional formula, CEZS, showed similar improvements to erythromycin on SNOT-20 and LM in a blinded study of CRS [36]. In an open label study in ARS, the decrease in VAS-nasal blockage in the oral CEZS group was not significantly different compared to the decrease in the amoxicillin group (Li MJ 2014a) [39]. Overall, the best available evidence for improvements in RS

symptoms appeared to be for LDXGT and modified CEZS, since both were tested in blinded studies as well as in open label studies. These formulae are recommended by Chinese text-books for people with RS and are prescribed according to syndrome differentiation [71].

## Clinically important differences

Assessments of baseline balance and calculation of within-group changes were conducted to determine the magnitude of change in each outcome and whether there were minimal clini-cally important differences (MCID). In a validation study for SNOT-20, a change of 0.8 points (16%) in the top five items was considered MCID [72]. In contrast, MCID values for SNOT-22 were based on summed total scores. These were 8.9 points (8.1%) [73] and 9.0 points (8.2%) [74]. Searches did not identify MCID for VAS-TNS, LM, LK, MTT or MTR. The within group changes for each of these outcomes are included in the Supporting information.

In this review, a 12-week study [46] that summed the five most severe SNOT-20 items (total 25 points) found a reduction of 6.00 points (24%) in the pharmacotherapy group, exceeding MCID; and a reduction of 6.33 points (25.3%) in the IM group indicating a small additional effect for adding oral LDXGT. Some studies summed the 20 items of SNOT-20 to provide a score out of 100. Since summed scores were used, as in SNOT-22, we selected 8.1% change in mean scores as the criterion for MCID. In the two placebo-controlled studies, reductions within test groups were 12.8% for LDXGT [43] and 6.9% for CEZS [36], suggesting the result for LDXGT appears clinically meaningful, while CEZS did not meet this MCID threshold. This result suggests that the best available evidence for CRS was for LDXGT followed by modi-fied CEZS. We did not assess possible MCID for open label studies since the lack of blinding may have led to inflation of effect sizes.

## Limitations

A limitation of this review is methodological weakness in some included studies. Of the 34 RCTs, 20 (58.8%) applied appropriate methods for sequence generation and five (14.7%) used placebos for blinding participants. However, allocation concealment was described in only three studies and none of the studies had locatable protocols. For the blinded comparisons between CHMs and pharmacotherapies, both groups tended to improve [36, 44]. However, it was not possible to determine whether the improvements in the open-label studies were due to the interventions, or a result of improved overall care associated with inclusion in a clinical trial. In IM studies, addition of the CHMs tended to provide additional benefits but the studies were not blinded, so this may have been a non-specific effect due to the provision of an addi-tional therapy. Statistical heterogeneity was evident in some pooled results, and most pools had fewer than 500 participants. These issues limited the meaningfulness of the pooled effect size estimates. Five CHMs were tested in multiple studies but none were tested in multiple blinded studies, so our confidence in the reliability of the evidence for these CHMs is limited. Although major safety issues were not found, some data were missing or incomplete.

## Conclusions

Strengths of this review included comprehensive searches, focus on internationally recognised outcome measures, assessments of baseline balance, and calculation of any minimal clinically important differences (MCID).

This review suggests that certain CHMs may have improved CRS symptoms, scores for sinus imaging, and measures of mucociliary clearance. Changes in SNOT-20 may have been clinically meaningful for LDXGT in one of the blinded studies. However, additional blinded RCTs of LDXGT and other CHMs are required to test whether these results can be replicated.

There were too few studies of ARS for any conclusions to be drawn. Further well-designed studies are required. Future IM studies require a placebo for the CHM in the control group to enable blinding. All future studies require adequate sample sizes, details of the quality control of the CHMs, complete safety reporting, and rigorous methodology detailed in a protocol that is available to reviewers.

## Supporting information

**S1 Checklist. PRISMA 2009 checklist.**
(DOC)

**S1 Table. Additional data.** Including: Databases that were searched and PubMed search terms for CHM for RS; List of excluded studies with reasons; Ingredients of the CHM interventions, manufacture and dosage used in the included studies and funding; Main ingredients of the Chinese herbal medicines; Risk of bias judgements for included studies; SNOT: Meta-analysis results for chronic rhinosinusitis at end of treatment and changes within treatment and control groups; SNOT-20-subscales: Meta-analysis results for CRS at end of treatment change within treatment groups and control groups; VAS-TNS Meta-analysis results for CRS at end of treatment and change within treatment and control groups; VAS-IS Meta-analysis results for CRS at end of treatment and change within treatment groups and control groups; LM Meta-analysis results for CRS at end of treatment and change within treatment and control groups; LK Meta-analysis results for CRS at end of treatment and change within treatment and control groups; MTT Meta-analysis results for CRS at end of treatment and change within treatment and control groups; MTR Meta-analysis results for CRS at end of treatment and change within treatment and control groups; SF-36 Meta-analysis results for CRS at end of treatment and change within treatment groups and control groups; and Details of reported adverse events from included studies.
(DOCX)

## Acknowledgments

We wish to thank Dr Iris WY Zhou and Dr Meaghan Coyle for their assistance with searches.

## Author Contributions

**Conceptualization:** Jing Cui, Wenmin Lin, Brian H. May, Christopher Worsnop, Anthony Lin Zhang, Xinfeng Guo, Chuanjian Lu, Charlie C. Xue.

**Data curation:** Jing Cui, Brian H. May, Qiulan Luo.

**Formal analysis:** Jing Cui, Wenmin Lin, Brian H. May, Qiulan Luo.

**Funding acquisition:** Chuanjian Lu, Charlie C. Xue.

**Investigation:** Jing Cui, Wenmin Lin, Brian H. May, Qiulan Luo, Anthony Lin Zhang.

**Methodology:** Wenmin Lin, Brian H. May, Anthony Lin Zhang, Xinfeng Guo, Chuanjian Lu, Charlie C. Xue.

**Project administration:** Anthony Lin Zhang, Xinfeng Guo, Chuanjian Lu, Yunying Li, Charlie C. Xue.

**Resources:** Anthony Lin Zhang, Xinfeng Guo, Yunying Li, Charlie C. Xue.

**Supervision:** Brian H. May, Christopher Worsnop, Anthony Lin Zhang, Xinfeng Guo, Chuanjian Lu, Yunying Li, Charlie C. Xue.

**Writing – original draft:** Jing Cui, Wenmin Lin, Brian H. May.

**Writing – review & editing:** Brian H. May, Qiulan Luo, Christopher Worsnop, Anthony Lin Zhang, Xinfeng Guo, Chuanjian Lu, Yunying Li, Charlie C. Xue.

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
