## [Decision Letter · Decision Letter 0]

17 Oct 2022

PONE-D-22-23604Chinese herbal therapy in the management of rhinosinusitis - A systematic review and meta-analysisPLOS ONE

Dear Dr. Charlie Changli Xue,

Thank you for submitting your manuscript to PLOS ONE. After careful consideration, we feel that it has merit but does not fully meet PLOS ONE’s publication criteria as it currently stands. Therefore, we invite you to submit a revised version of the manuscript that addresses the points raised during the review process.

We look forward to receiving your revised manuscript.

Kind regards,

Huijuan Cao, Ph.D.

Academic Editor

PLOS ONE

Journal Requirements:

"None of the authors were involved in the clinical studies included in this review. None of the authors have a conflict of interest."

6. Please upload a copy of Figure 6, to which you refer in your text on page 16. If the figure is no longer to be included as part of the submission please remove all reference to it within the text.

Reviewers' comments:

Reviewer's Responses to Questions

**Comments to the Author**

1. Is the manuscript technically sound, and do the data support the conclusions?

Reviewer #1: Yes

Reviewer #2: Partly

2. Has the statistical analysis been performed appropriately and rigorously? 

Reviewer #1: Yes

Reviewer #2: Yes

3. Have the authors made all data underlying the findings in their manuscript fully available?

Reviewer #1: Yes

Reviewer #2: Yes

4. Is the manuscript presented in an intelligible fashion and written in standard English?

Reviewer #1: Yes

Reviewer #2: Yes

5. Review Comments to the Author

Reviewer #1: Dear Authors,

Thank you for conducting this study entitled "Chinese herbal therapy in the management of rhinosinusitis - A systematic review and meta-analysis" for possible publication in the esteemed journal "PLOS ONE". I have two comments:

1. The descriptive part of the introduction section is deficient. It needs in more detail, for example the history of CHM in the treatment of rhinosinusitis.

2. The authors should add the starting time in searching the studies.

Reviewer #2: This is a well-designed systematic review which assessed the therapeutic effective of Chinese herbal medicine for rhinosinusitis. I only have one suggestion for the authors, as a systeamtic review, it's very important to provide the current best evidence for a research question which help the clinical practitioners to make the decision. Thus, the quality of the evidence is quite important. I suggest the authors to use GRADE criteria to evaluate the overall quality of the evidence and present the results with summary of finding tables.

6. PLOS authors have the option to publish the peer review history of their article (what does this mean?). If published, this will include your full peer review and any attached files.

Reviewer #1: **Yes: **Raid M. Al-Ani

Reviewer #2: **Yes: **Huijuan Cao

---

## [Author Response · Author response to Decision Letter 0]

8 Nov 2022

Response to Reviewers

We wish to thank the reviewers for their constructive comments. We have made changes to the MS accordingly. These are outlined below and highlighted in the MS using track changes.

Reviewer #1: Dear Authors,

Thank you for conducting this study entitled "Chinese herbal therapy in the management of rhinosinusitis - A systematic review and meta-analysis" for possible publication in the esteemed journal "PLOS ONE". I have two comments:

1. The descriptive part of the introduction section is deficient. It needs in more detail, for example the history of CHM in the treatment of rhinosinusitis.

Response:

We thank the reviewer for this comment. In the introduction section we have added a passage on history (MS page 3) with two references added. 

17. May BH, Lin W. Ch 3. Classical Chinese medicine literature. In: Xue CC, Lu C, Zhang AL, Guo X, Wen Z, editors. Evidence-based Clinical Chinese Medicine Volume 25: Rhinosinusitis. 25. Singapore: World Scientific Publishing Co; 2022.

18. Li YY, Liu SP. Otorhinolaryngology: clinical diagnosis and treatment in Chinese medicine. Beijing: Peoples Medical Publishing House; 2013.

2. The authors should add the starting time in searching the studies.

Response:

We thank the reviewer for this comment. We have provided additional information in the section on Information sources and search strategy.

We have amended the search strategy accordingly. See MS page 6 and supporting information S1 Table.

Reviewer #2: This is a well-designed systematic review which assessed the therapeutic effective of Chinese herbal medicine for rhinosinusitis. I only have one suggestion for the authors, as a systematic review, it's very important to provide the current best evidence for a research question which help the clinical practitioners to make the decision. Thus, the quality of the evidence is quite important. I suggest the authors to use GRADE criteria to evaluate the overall quality of the evidence and present the results with summary of finding tables.

Response:

We thank the reviewer for this suggestion. We have: 

• added references to GRADE in the Data analysis section (page 6), and

• conducted a series of GRADE assessments and provided a series of tables. These have been placed in a separate section following Adverse events (Page 20-25). 

• We also have mentioned the GRADE assessments in the discussion section (page 26).

• We have added two references relating to GRADE (page 7)

34. Schünemann H, Brozek J, Guyatt G, Oxman A, editors. GRADE handbook for grading quality of evidence and strength of recommendations (The GRADE Working Group): Retrieved from http://www.guidelinedevelopment.org/handbook/ (Jan 2017); 2013.

35. Schünemann HJ, Higgins JPT, Vist GE, Glasziou PP, Akl EA, Skoetz N, et al. Chapter 14: Completing ‘Summary of findings’ tables and grading the certainty of the evidence. In: Higgins JPT, Thomas J, Chandler J, Cumpston M, Li T, Page MJ, et al., editors. Cochrane Handbook for Systematic Reviews of Interventions version 60 (updated July 2019) Available from www.training.cochrane.org/handbook: Cochrane; 2019.

Additional changes

We have checked the manuscript and supporting information carefully and made minor amendments in track changes. In one place there was a numerical error which has been corrected (page 14). The numbers were correct in Figure 3 and this change does not affect the overall result. 

In the Manuscript version the references have all been formatted according to Plos One.

We have provided a track changes version of the Supporting information file.

We believe that these changes have improved the paper and wish to thank the reviewers for their comments and suggestions.

---

## [Editor Report · Decision Letter 1]

17 Nov 2022

Chinese herbal therapy in the management of rhinosinusitis - A systematic review and meta-analysis

PONE-D-22-23604R1

Dear Dr. Xue,

We’re pleased to inform you that your manuscript has been judged scientifically suitable for publication and will be formally accepted for publication once it meets all outstanding technical requirements.

Kind regards,

Huijuan Cao, Ph.D.

Academic Editor

PLOS ONE

---

## [Editor Report · Acceptance letter]

21 Nov 2022

PONE-D-22-23604R1 

Chinese herbal therapy in the management of rhinosinusitis - A systematic review and meta-analysis  

Dear Dr. Xue:

I'm pleased to inform you that your manuscript has been deemed suitable for publication in PLOS ONE. Congratulations! Your manuscript is now with our production department. 

Kind regards, 

on behalf of

Dr. Huijuan Cao 

Academic Editor

PLOS ONE